# Physical-Mechanical Properties of Peat Moss (Sphagnum) Insulation Panels with Bio-Based Adhesives

**DOI:** 10.3390/ma15093299

**Published:** 2022-05-04

**Authors:** Marco Claudius Morandini, Günther Kain, Jonas Eckardt, Alexander Petutschnigg, Jan Tippner

**Affiliations:** 1Department for Forest Products Technology and Timber Construction, Salzburg University of Applied Sciences, Markt 136a, 5431 Kuchl, Austria; gkain.lba@fh-salzburg.ac.at (G.K.); jonas.eckardt@fh-salzburg.ac.at (J.E.); alexander.petutschnigg@fh-salzburg.ac.at (A.P.); 2Salzburg Center for Smart Materials, c/o Department of Chemistry and Physics of Materials, Paris Lodron University of Salzburg, Jakob-Haringer-Strasse 2a, 5020 Salzburg, Austria; 3Department of Wood Science and Technology, Mendel University, Zemědělská 3, 61300 Brno, Czech Republic; tippner@centrum.cz; 4Department for Wood Restoration Technology, Higher Technical College Hallstatt, Lahnstraße 69, 4830 Hallstatt, Austria

**Keywords:** peat moss, insulator, tannin, animal glue, bio-based materials, bio-based adhesives

## Abstract

Rising energy and raw material prices, dwindling resources, increased recycling, and the need for sustainable management have led to growth in the smart materials sector. In recent years, the importance and diversity of bio-based adhesives for industrial applications has grown steadily. This article focuses on the production and characterization of insulation panels consisting of peat moss and two bio-based adhesives. The panels were pressed with tannin and animal-based resins and compared to panels bonded with urea formaldehyde. The physical–mechanical properties, namely, thermal conductivity (TC), water vapor diffusion resistance, modulus of rupture (MOR), modulus of elasticity (MOE), internal bond (IB), compression resistance (CR), water absorption (WA) and thickness swelling (TS) were measured and analyzed. The results show that the insulation effectiveness and mechanical stability of moss panels bound with tannin and animal glue are comparable to standard adhesives used in the composite industry.

## 1. Introduction

According to Zarestky and Collins (2017), global challenges such as climate change, limited resources and political instability are some of the reasons why many organizations and companies face increasing pressure to focus on the resource-related aspects of producing or procuring their products [1]. In the last few years, resource usage, resource efficiency and sustainability have been targeted in the environmentally-related political incentive plans of the European Union (EU). In these incentive plans, the EU concentrates on five main categories of resources: replenishing materials, the development of new technologies for efficient energy consumption, water, land and the reduction of CO_2_ emissions [2].

New materials are crucial for securing and expanding the competitive positioning of developed industrial countries. They play a key role in system innovations and are vital for technological and economic progress. Their use often determines the performance and degree of innovation of future-oriented technologies. [3]

The variety of combinations and the production of composites have been revolutionized by the acceptance and diversity of biomaterials. The development of new technologies has been greatly advanced by this in recent years [4,5]. The concept of green composites is well studied and there has been major progress in the last decade. The reinforcing material and the matrix/resin are key components that determine the properties of composite materials. Accordingly, green composite reinforcements are usually natural polymer or biomass products and the matrices/resins are waste polymers. Environmentally-friendly green composites have experienced an increase in production in recent years and have found wide application in various areas. Materialistic and technological considerations must also be considered in order to improve the cost-effectiveness of producing modern green composites. Overall, green composites are making great progress and are becoming a promising material for most purposes in the near future.

Biogenic raw materials and components will play an important role in the future to replace or minimize the use of petroleum-based materials. In the development of new materials, the focus is on independence from raw fossil materials and meeting the requirements of a bio-based society. Thus, resources that have been left unused so far could play a key role [6,7].

The cultivation of peat moss (*Sphagnum*) has been investigated for several years already [8]. Globally, moors contain twice as much carbon as the biomass of all forests, although they only cover 3% of the world’s area [8,9]. Peatlands that have not been sufficiently wet for a long time due to drainage projects, or that have already been degraded can be recultivated and used for agriculture [8,10]. Due to the unique properties of sphagnum moss, it can be sustainably harvested every 4 years [11].

About 3–6 tons of peat moss can be harvested per hectare per year. This value is related to the dry mass of the raw material [8,9]. In order to exploit the full potential of its carbon uptake, it is important to ensure a high-water level in the cultivated areas. In Germany, agriculturally-used peatlands emit approximately 40 million tons of CO_2_ [9]. The determining factors for the thermal conductivity (TC) of moss panels include the moisture content, the density and the dimensions of the panel [8,12]. The higher equilibrium humidity is due to the special cell structure of the moss, which enables the plant’s high-water storage capacity [13]. Due to their large, water-retaining hyaline cells (Figure 1), sphagnum mosses are able to store large amounts of water. Peat moss can be dried and then stored in water until maximum swelling occurs. In the process, the moss is able to absorb an average of 14 times its dry weight in water and a significant increase in volume occurs [11,14]. 

The insulating effect of peat moss panels in the building envelope has already been the subject of several studies. The results showed that in a standard climate (65% humidity and 20 °C temperature), an equilibrium moisture content (EMC) of 18% is established. Different gluing systems based on bio-based adhesives have been used for the production of different panels [8]. The thermal conductivity of moss mats with a density of 100 to 250 kg/m^3^ ranges from 0.039 to 0.065 W/mK [12,15]. The untreated moss at its original density has a higher thermal conductivity than boards with a density of 100 kg/m^3^ [16]. This effect can be described by the effects of convection and radiation at different temperatures. In this context, the panel structure has the greatest effect on TC [17]. If peat moss is compared with other systems or insulation materials available in the market, moss can be placed in the range of mineral wool [14]. The TC accounts for 0.35 mW/mK. This is significantly lower than other insulation materials based on renewable resources [9]. 

Moss panels or mats have to be bound with some kind of resin. UF resin has been shown to be a suitable binding system, but the question of more sustainable alternatives remains. Tannins naturally occur in the vegetable world and they are therefore extracted from the bark and wood of certain trees as well as from grape and other fruit skins [18]. Traditionally, they have been used in the tanning industry. As shown in previous articles, tannins can also be used in the production of bio-based wood adhesives [16]. Research on tannin-based adhesives was first reported in the 1950s and has remained of interest ever since [17]. Such adhesives have been proven to be suitable for industrial usage as a wood adhesive by several producers [19]. A very common approach is to use hexamine as a hardener but the use of furfuryl alcohol has also been shown to be suitable for the manufacturing of particle boards [14,16]. The latter can be produced from saccharidic resources [18]. The resin formulation used in the current study was developed and applied during pre-trials based on research focusing on tannin–furanic foams [18]. Tannin–furanic foams are known to be brittle, which can be compensated for by adding plasticizers to the formulation [20]. According to the literature, plasticizers have proved to be suitable for the tannin–furanic system, namely, (poly)ethylene glycol and glycerol [21,22]. Ethylene glycol is also supposed to allow better solubilization of the tannin [23]. Animal glue is a collagen-based material that is readily available and has been widely used throughout history as an adhesive, binder and additive for plasters and mortars [24]. It is derived from hides and consists of long protein molecules that are made up of amino acids linked by covalent peptide bonds [25].

The results of recent research on peat moss confirm that this material has very good isolation properties [16,19]. For this reason, the use of biologically degradable adhesives is being investigated in this context in order to improve the overall ecological performance of such products. Within the framework of this investigation, the production of panels produced with tannin and animal glue is discussed. Their mechanical–physical properties are discussed compared to panels bound with urea formaldehyde.

## 2. Materials and Methods

The raw peat moss was provided by the German company Paludi Culture Ramsloh. The material was conditioned as specified in ÖNORM EN ISO 18134-2 [17].

### 2.1. Chemicals and Reagents

Industrial mimosa tannin extract, Weibul AQ was supplied by Tanac S.A. (Montenegro, Brazil). Furfuryl alcohol (98% technical) was provided by Transfurans Chemicals (Geel, Belgium). Glycerol (99.5% p.a., anhydrous) and sulfuric acid were purchased from Roth (Karlsruhe, Germany), and diethylene glycol (≥99%, for synthesis) was purchased from Merck (Darmstadt, Germany). The animal glue used for the production of the panels is based on fish skin and was commercially supplied by Alpina KG (Vienna, Austria). Urea formaldehyde with a solid content of 66% from Metadynea (Krems, Austria) was used.

### 2.2. Preparation of Resins

The mass fraction of the different chemicals used in the formulation of the tannin resin can be seen in Table 1. At first, the different liquid parts of the formulation, furfuryl alcohol, glycerol, water and diethylene glycol were mixed before adding the tannin. Once the tannin was added to the mixture, it was stirred for about 1 min until homogenization. The stirring speed was set at a variable speed between 700–1000 rpm. In the last step, the sulfuric acid with a concentration of 2 mol/L was added as a catalyst, and the mixture was stirred for another minute before obtaining the resin, which was ready for processing. To ensure that the glue could be applied homogeneously, the viscosity of the animal glue had to be adjusted. For this, the fish glue was diluted with 20% reverse osmosis water. After stirring, the viscosity of the animal glue was between 60 and 80 mPa.s. The osmosis water had a temperature of 27.5 degrees. The basic viscosity of urea formaldehyde resin, Prefere 10F102 (66% solid content) was 60–90 mPa.s. The tannin resin had a viscosity of 40–70 mPa.s with a solid content of 50%. These glue properties were suitable to achieve homogeneous gluing.

### 2.3. Production of the Peat Moss Panels 

The calculations for the binder and raw material content correspond to the current state-of-the-art in the wood-based materials industry. The conditioning cycle is an essential factor for obtaining a representative board density. Therefore, the particles were exposed to a standard climate of 65% humidity and 22.5 °C temperature. Since the peat moss can absorb about 30 times its own weight in moisture, the particle size, the viscosity of the binder and the gluing method are particularly important to achieve adequate results. To achieve an even distribution of the adhesive, the material was crushed manually. The fractional composition of the crushed material ranged from 1 to 3 mm. 

In order to obtain a homogeneous panel, the particles were crushed in the ploughshare mixer for approximately 10 min. Figure 2 shows a microscopic image (SEM) of the crushed material. Scanning electron microscopy (SEM) images were taken using a Zeiss Ultra Plus field emission scanning electron microscope equipped with an annular backscatter electron detector (Carl Zeiss AG, Oberkochen, Germany). The acceleration voltage was set to 5 kV and the working distance was adjusted to between 4 and 6 mm. Prior to the imaging, the samples were coated with a thin layer of gold using a sputter coater with a current of 40 mA and coating time of 60 s.

A ploughshare mixer (ETM-WHB75m, amixon GmbH, Berlin, Germany) was used to mix the individual components of the fiber mat. The following adhesives, industrial mimosa tannin extract-based glue, animal glue and urea formaldehyde as described above, were used to bind the particles. In the first step, the moss fibers were filled into the drum of the mixer. 

In order to ensure the even distribution of the binder, the adhesive was evenly sprayed into the drum mixer using compressed air. The injection pressure was set to 1.5 bar during the entire process. Until further processing of the resinated moss, the fibers were stored at a temperature of 10 °C to slow down the hardening of the resin. Afterwards the leaded particles were placed in press molds and were pressed with a Höfer (Taiskirchen, Austria) HLOP 280 laboratory press. A pressing temperature of 180 °C was used for the panels with tannin-based glue and UF. The panels with animal glue were produced at a temperature of 60 °C. 

According to the design of the experiment (Table 2), panels with densities between 100–250 kg/m^3^ were produced. The critical point of the processability was the moisture of the particles. In order to allow the moisture to escape continuously over the entire cross-section of the plate, a total of three deaeration phases had to be used in the press cycle. After formatting, the boards (Figure 2) measured 32 × 32 × 2 cm.

### 2.4. Physical–Mechanical Characterization

All panels were tested for their mechanical properties. The tests were conducted according to the procedure specified in the European standards EN 310, EN 317, EN 319 (European Committee for Standardization [CEN] 1993a, 1993b, 1993c), DIN 52192 (DIN 1979b), and DIN 52188 (DIN 1979a). After conditioning the panels at 20 °C and 65 percent relative air humidity for one week, the samples were cut to a specimen size according to EN 326-1 (CEN1994). The specimens’ static bending (MOR and MOE) characteristics were determined according to ÖNORM EN 12089 [19,20,21] and the internal bond (IB) according to ÖNORM EN 319 [22]. The other properties that were tested were compression resistance (CR), thickness swell (TS) and water absorption (WA) after 24 h of immersion in water at 20 °C.

The vapor diffusion resistance (VDR) was determined according to ÖNORM EN ISO 12572 [21]. For this study, the different boards were cut into cylinders. The samples were then placed in a plastic container with a diameter of 100 mm. For the dry cup test, the plastic container was filled with silica gel and for the wet cup test with osmosis water. The filling height was 15 mm below the specimen in both cases (Figure 3). 

These materials create an air humidity of 0 and 100%, respectively. The cups with the samples were stored in an air-conditioned chamber (20 °C, 65% RH). Periodical weighing enabled us to determine the vapor stream per time unit, which is proportional to the diffusion resistance.

### 2.5. Statistical Analysis

The statistical analysis of the data was performed using the software SPSS 18 (IBM). The linear relationships between the dependent and independent variables were quantified by applying regression analysis. The effects of nominal factors were evaluated using an analysis of covariance (ANCOVA), whereby the bulk density was considered to be the covariate factor. In order to estimate the explanatory power of the independent variables on the dependent variables, partial eta squared values ($\eta^{2}$) were calculated. The application of ANCOVA requires both variance homogeneity and normally distributed data [23], a chi-square test for deviation from the normal distribution was performed in each case beforehand. In addition, the mean values with the corresponding standard deviations of the individual results were calculated. 

## 3. Results and Discussion

The panels had an equilibrium moss moisture content of 10.4 ± 0.5% at a density of 100 kg/m^3^, 10.6 ± 0.3% at a density of 150 kg/m^3^ and 11.1 ± 0.7% at a density of 250 kg/m^3^ after conditioning them at 20 °C and 65% relative air humidity. 

### 3.1. MOR/MOE

The flexural strength of the tested moss panels ranged from 0.03 to 1.45 N/mm^2^ (Figure 3). The empirical model (*p* = 0.000) showed that 95.6% of the differences in flexural strength can be explained by density and resin type. The effects of density and resin type are significant (*p* = 0.001). The explanatory power of the panel density is the highest; it explains 53.1% of the variation in the modulus of rupture (MOR). The MOR increases 0.7 N/mm^2^ by increasing the density 100 kg/m^3^ (Figure 4).

The results of the statistical evaluation regarding the bending stiffness are also significant (*p* = 0.000) and describe 96.1% of the differences in the modulus of elasticity. The adhesive type and density are also determining factors. Both factors are highly significant (*p* = 0.000) with an explanatory power of 89% and 54%, respectively. The stiffness of the panels increases 69 N/mm^2^ per 100 kg/m^3^ increase in density (Figure 5).

Compared to previous studies with materials such as wood particles without adhesive, 100–300 kg/m^3^, MOR 0–2 N/mm^2^ [25]; and bagasse particleboard bound with citric acid and sucrose, 300–500 kg/m^3^, MOR 1–7 N/mm^2^ [25], it can be seen that the peat moss panels with bio-based adhesives show a similar flexural strength range. In this study, the MOR of the panels with animal glue is on average 10% higher and the MOR of tannin panels is 10% higher than that of the UF panels. This contradicts other findings (e.g., bark and normal wood fibers) that UF has a rather good gluing efficiency. Considering the high variation in the MOR of UF panels (Figure 5), the moss resin interaction should be a topic of further research. It might be that the high hygroscopicity of peat moss adversely affects the UF binding. Both the MOR and MOE show high variation with a panel density of 250 kg/m^3^, whilst the variation is low with the lighter panels. This suggests that the resin distribution is not optimal. Uneven resin distribution shows stronger adverse effects with higher densification, which would explain the current findings. Further research will have to be conducted to study the quality of resin distribution and its optimization.

A previous study on the load-bearing capacity of an insulation board (250–1300 kg/m^3^) made of miscanthus bound with mineral binders, obtained a flexural strength of 1.5 N/mm^2^ for the lighter boards [26]—the same as the moss samples. The MOE of light bark insulation boards with a density of 250 kg/m^3^ is around 50 N/mm^2^ [27], a parameter that was surpassed by the peat moss boards with the same density. Although the moss panels with bio-based adhesives are sufficiently stiff and strong at densities above 150 kg/m^3^, the results for the panels with urea formaldehyde are lower at densities of 100 and 250 kg/m^2^. On average, the MOE of tannin-bound panels is 1% higher than the MOE of panels with animal glue and 34% higher than that of the panels with UF resin. Tannin hardens as a crystalline matrix [28], which is most probably the reason for the increased stiffness.

### 3.2. Internal Bond

The internal bond varied between 0.006 and 0.036 N/mm^2^. The peat moss panels with the animal glue showed the highest IB, and 76% of the variation in the IB can be attributed to the different adhesive types. This effect is statistically significant (*p* = 0.007). Focusing on panels with a density of 250 kg/m^3^ (Figure 6), the IB of the animal glue-bonded boards is higher than that of the boards with tannin-based adhesives. The mats with animal glue had the highest IB of 0.027 N/mm^2^ at a density of 150 kg/m^3^ (Figure 6).

Similar to wood-based panels, the IB increases with density. Whilst the IB of low-density moss panels shows varying characteristics with different resins, the animal-glue and the tannin-bound panels seem to be advantageous with increasing density. This can be attributed to the fact that tannin resin hardens to a hard, crystalline matrix and its lower initial viscosity, compared to UF, favors better resin distribution. 

The internal bond of other insulation products varies from 0.003 to 0.05 N/mm^2^ [29], indicating that the proposed peat moss insulation boards have reasonable IB values. Compared with urea formaldehyde, the specific differences between the resins show that the bio-based adhesives have high bonding efficiency at higher board density. 

### 3.3. Compression Resistance (CR)

The compression resistance of the investigated samples ranges from 0.093 (M1) to 0.923 N/mm^2^ (AG3) (Figure 7). Almost all (99.9%) of this variation is related to differing resin type (*p* = 0.001)) and density (*p* = 0.010). The explanatory power of the resin type is 98% significantly higher than that of the densities with 84%.

The CR of panels with animal glue and tannin-based adhesives with a density of 100/kgm^3^ are comparable. At a density of 250 kg/m^3^, the panels that were produced with the animal glue are 50% more pressure-resistant than their counterparts produced with tannin-based glue. It should be noted that the CR of the panels with bio-based glue and the urea formaldehyde were tested at 10% compression. 

Wood fiber panels have a compression resistance of 0.04 to 0.2 N/mm^2^ [28], indicating that peat moss panels have adequate pressure resistance for insulation applications. Insulation panels made from miscanthus and a mineral binder showed a CR of 0.3 N/mm^2^ with a density of 250 kg/m^3^ [26], which confirms the competitiveness of the peat moss panels.

In general, the mechanical properties of peat moss panels are superior to those of wood-based panels at the same density, most likely due to the high compression ratio of the moss panels. The compression ratio for moss panels with a density from 50 to 250 kg/m^3^ is between 3.8 and 19 [9].

### 3.4. Water Absorption and Thickness Swell

The water absorption ranges from 7–38 kg/m^2^, with the lowest WA in the tannin-based panels and the highest WA in the heavy panels with animal glue. (Figure 8 and Figure 9). The effect of the resin (*p* = 0.000) is statistically significant with an explanatory power of 96%.

The water absorption increases with density. At a density of 100 kg/m^3^, the WA of different specimens is comparable and below 18%. At higher densities, the water absorption is higher in moss mats with animal glue than in the UF and tannin-bound samples. The latter showed a maximum water absorption of 20% at 250 kg/m^3^.

A similar coherence was investigated for TS, which has a variation of 99.2%, caused by the panel type (*p* = 0.000) and the density (*p* = 0.006). The explanatory power of the panel type (98%) exceeds that of density (34%). The TS amounts range from −17 to 120%. The light moss mats shrink when wetted [8]. At a density of 100 kg/m^3^, they swell 5%, which is comparable to resin-bound panels at 100 kg/m^3^. Up to a density of 150 kg/m^3^, all investigated samples showed a steep increase in TS with increasing density (not the moss mats with animal glue, which collapse at low densities when they are wetted). At higher densities, the animal glue-based samples swell to disproportionately high levels of 120%, whereas the tannin-bound panels show a lower increase in TS.

In comparing the boards bound with tannin and animal glue, it is clear that the tannin enhances the hydrophobic properties of the boards, as the WA and TS of the former is significantly lower. Tannin-bound panels show an even lower water absorption than UF-bound panels, and the TS is comparable. This advantage may be put into perspective in ongoing research, as the amount of resin needs to be decreased in an industrial context for economic efficiency reasons. Cotton batts absorbed 12–13, hemp 4.2, and EPS 1.5 to 3 kg/m^2^ of water in an impregnation test [29]. 

The tannin-bonded peat moss mats are comparable to cotton, and the wet-processed sheets showed significantly higher water absorption. The high-water absorption of sphagnum moss is caused by the water-retaining hyaline cells of sphagnum moss, which are optimized in nature to absorb huge amounts of water and could be optimized further by adding hydrophobic agents [25]. The thickness swelling of peat moss samples is high compared to a study that found a TS of 13–20% for rice straw panels bonded with polymeric methylene diphenyl diisocyanate (pMDI) at a density of 250 kg/m^3^ [26]. In a study on the development of insulation boards made from cat-tail, the TS was limited to 15% after 24 h of water storage [9,28].

The hydrophobic character of moss is disadvantageous for building products with low hygroscopicity. There might be potential for capillary active interior insulation layers, where temporarily occurring condensation water can be absorbed and dries out through the insulation layer. This process would be supported by the moss panels’ low water vapor diffusion resistance.

### 3.5. Thermal Conductivity (TC)

The TC of the tested samples ranges between 0.039 W/mK and 0.065 W/mK (Figure 10). This scatter is significantly (*p* = 0.001) affected by the sample density, while the effect of the different resins is not significant (*p* = 0.201). A significant percentage, i.e., 77% (*p* = 0.000), of the variation in TC can be explained by density (Figure 10). The lowest TC of 0.04 W/mK was measured at 100 kg/m^3^, from where the TC increases by 0.11 W/mK per 100 kg/m^3^ increase in density. 

The moss specimens with a TC between 0.038 and 0.065 W/mK, depending on density, are similar to natural insulation materials such as wood fibers or flax and synthetic materials such as expanded polystyrene. The findings are in accordance with previous studies on peat moss insulation applications. [9,16] In the cited studies, the optimization of density was broadly discussed and showed that the lowest TC was achieved with a moss mat density of approximately 50 kg/m. Due to stability reasons, the density of the current panels was set much higher, showing adequately low TC.

### 3.6. Vapor Diffusion Resistance (VDR)

The test procedure for determining the water vapor diffusion resistance of the individual board samples was carried out over a period of 27 days. With regard to the calculation method, it should be noted that the DIN ISO 12572 [21] standard requires the inclusion of the water vapor diffusion resistance of the air layer between the specimen and the test substance for materials with an Sd value of less than 0.2 m. The water vapor diffusion resistance of the specimen was determined in the test. As it can be seen in Figure 11 he impact of the test type (dry/wet) is significant (*p* = 0.001), and that of the density is also significant (*p* = 0.34) (Figure 11). If the VDR is examined in the dry test, the density increases by 1.9/100 kg/m^3^. The R^2^ of the density and VDR in this case is 0.91. The wet test presents a different picture with an R^2^ of 0.74 between density and VDR, but a rather constant VDR value.

Compared to the peat moss boards bonded with urea formaldehyde, the bio-based adhesives showed a lower vapor diffusion resistance. The panels with tannin-based adhesives delivered the lowest values of all the insulation boards in the dry cup and wet cup test. For all boards and adhesive types, however, the high adhesive content is likely to have a significant influence on the results [30]. This also applies to insulation materials with high density. [30] According to other studies, the VDR value of peat moss boards with bio-based adhesives is lower than that of expanded polystyrene foam (30–35 VDR) [9,30,31]. For densities higher than 350 kg/m^3^ a linear relation exists between the two characteristics [31]. Referring to panels with the same density, the VDR is on average 90% lower in the wet cup test than in the dry cup test.

## 4. Conclusions

The results show that bio-based adhesives can be used as an alternative to conventional adhesives and suggest a wide range of applications for this material. The tannin-based resin and the animal glue were compared with urea formaldehyde for potential use in binding peat moss insulation panels. With regard to the MOR, IB, TS and WA, VDR, TC the tannin-based panels showed the best results. The lower-density boards have low mechanical stability. This limits their application to non-structural situations. In comparison with the results obtained for urea formaldehyde, lower vapor diffusion resistance was observed in both the resins based on natural resources, while the adhesive content remained the same. The peat moss samples have a low thermal conductivity that is comparable to that of other lightweight insulation materials. Their moisture absorption and water absorption properties limit their application to interior spaces or shaded areas. 

Finally, resin formulations and press parameters can be further optimized in the context of tannin-bound peat moss insulation boards. This is especially pertinent for industrial applications because a reduction in resin amount and press time directly influences the profitability of production.

## Figures and Tables

**Figure 1 materials-15-03299-f001:**
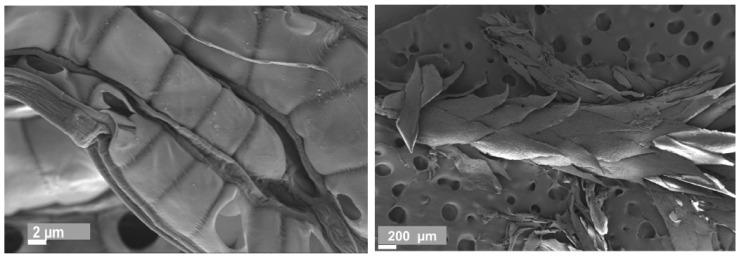
SEM image of dry peat moss at a resolution of 2 and 200 microns.

**Figure 2 materials-15-03299-f002:**
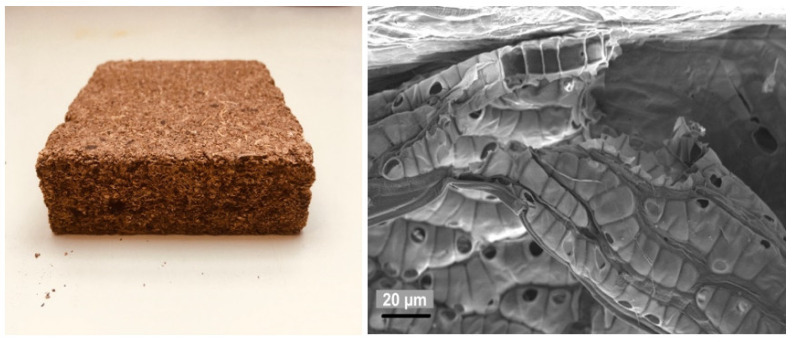
**Left** Peat moss panel bound with tannin-based adhesive, density 150 kg/m^3^; on the **right**, an SEM image of the peat moss at a scale of 20 µm.

**Figure 3 materials-15-03299-f003:**
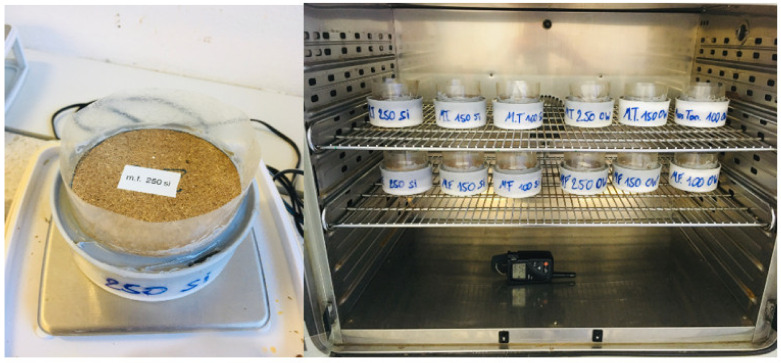
Embedded specimen (dry cup/wet cup).

**Figure 4 materials-15-03299-f004:**
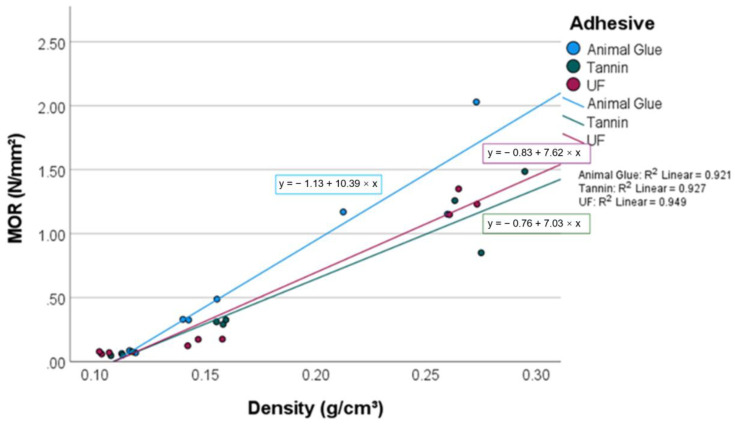
MOR of the investigated panels.

**Figure 5 materials-15-03299-f005:**
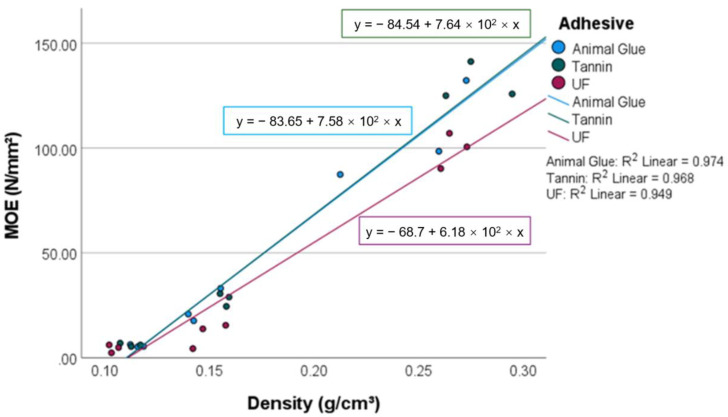
MOE of the investigated panels.

**Figure 6 materials-15-03299-f006:**
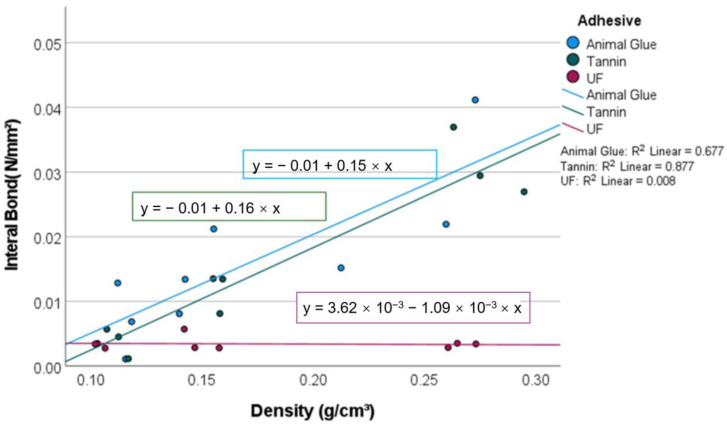
IB of the investigated panels (5 samples per category).

**Figure 7 materials-15-03299-f007:**
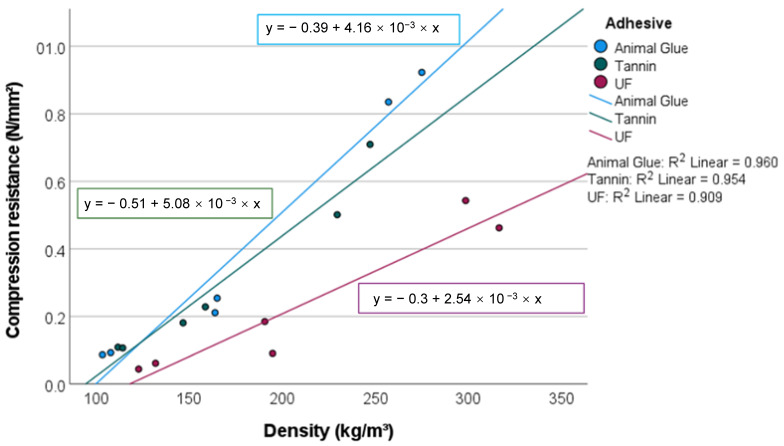
Compression resistance of the investigated panels.

**Figure 8 materials-15-03299-f008:**
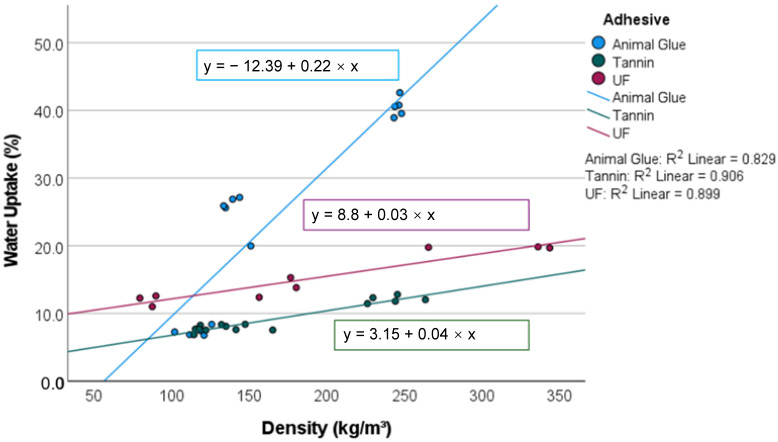
Water uptake of the investigated panels (5 samples per category).

**Figure 9 materials-15-03299-f009:**
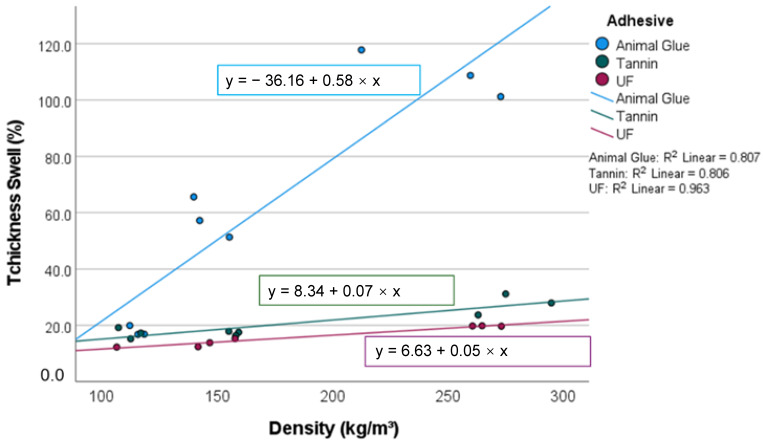
Thickness swell of the investigated panels (5 samples per category).

**Figure 10 materials-15-03299-f010:**
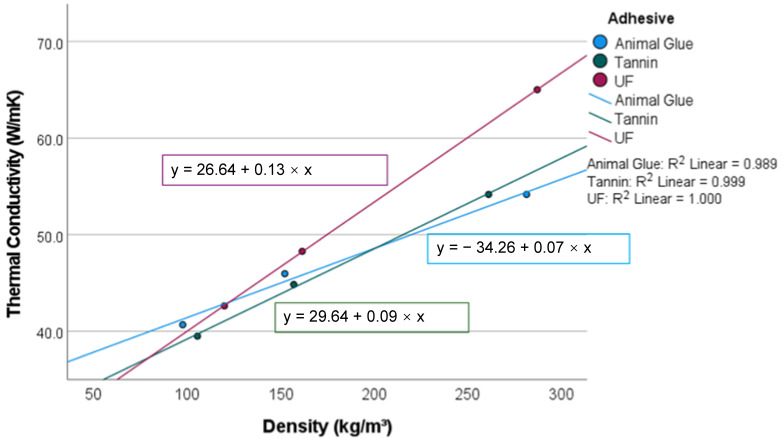
Thermal conductivity dependent on density.

**Figure 11 materials-15-03299-f011:**
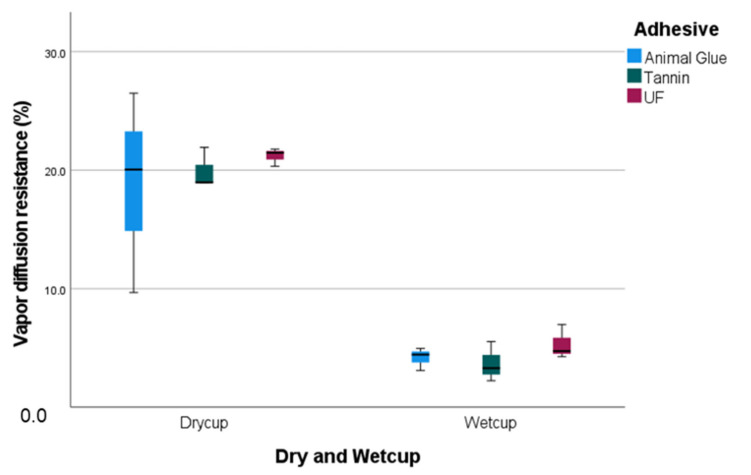
Vapor diffusion resistance of peat moss panels bounded with bio-based adhesives.

**Table 1 materials-15-03299-t001:** Formulation of the tannin resin (based on the oven-dry-mass).

Tannin	Furfuryl Alcohol	Glycerol	Water	Diethylene Glycol	Sulfuric Acid (2 mol/L)
43.9% (87.8 g)	25.4% (50 g)	8.7% (16 g)	5.6% (12 g)	4.8% (10 g)	11.6% (20 g)

**Table 2 materials-15-03299-t002:** Design of experiment for the production of peat moss panels.

Panel Type	Resin Content (%)	Target Density (kg/m^3^)	Replicates
AG 1	20	100	3
AG 2	20	150	3
AG 3	20	250	3
TA 1	20	100	3
TA 2	20	150	3
TA 3	20	250	3
UF 1	20	100	3
UF 2	20	150	3
UF 3	20	250	3

AG = panel bound with animal glue, TA = panel bound with tannin adhesive, UF = panel bound with urea formaldehyde.

## Data Availability

Not applicable.

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
