# Peer review of "Physical-Mechanical Properties of Peat Moss (Sphagnum) Insulation Panels with Bio-Based Adhesives"

_materials, 2022, doi:10.3390/ma15093299_

Round 1

Reviewer 1 Report

Dear authors,

The current work deals with a very interesting and novel topic such as the use of moss peat fibers and the possibily of incorporating tannin based glue to produce insulating panels. On average the quality of the work is good and I saw that it is the continuation of a previous work already published in this same journal during this year. However, I feel that the manuscript still present some points that should be revised and corrected before being considered for publication in this journal:

Title: No comments in this respect

Abstract: Well-structured, clear idea of the work done and within the word limitation demanded by the journal. However I detected that the half of the sentences in the abstract were the equal to those present in another work published in this same journal during this year (Production and Physical–Mechanical Characterization of Peat Moss (Sphagnum) Insulation Panels-doi.org/10.3390/ma14216601)

Previous article: Peat moss (sphagnum) is a commonly used sealant, fill, and insulation material in the past. During the efforts to rewet drained moors due to ecological considerations, the technical use of peat moss (sphagnum farming) again became the focus of attention. […] The specimens’ thermal conductivity, water vapor diffusion resistance, modulus of rupture, modulus of elasticity, internal bond, compression resistance, water absorption, and thickness swelling were measured. Physical–mechanical properties were adequate with the resin-bound panels, but not with wet process panels.

Current work: Peat moss (Sphagnum) was a commonly used sealant, fill and insulation material in the past. During the efforts to rewet drained moors due to ecological considerations, the technical use of peat moss (Sphagnum farming) is in the focus of attention again. […] The specimens’ thermal conductivity, water vapor diffusion resistance, modulus of rupture, modulus of elasticity, internal bond, compression resistance, water absorption and thickness swelling were measured. Physical-mechanical properties are adequate with the resin bound insulation 23 panels.

I strongly suggest that the authors reformulate the abstract to avoid plagiarism of their own previous article.

Keywords: I would suggest replacing “insulation material” by “insulator” and “renewable materials” by “bio-based materials”

Introduction: In general, the ideas and the background of the current work were adequately presented. However, the following points should be considered:

-Same issue detected in abstract section was found here. The text presented in pages 1-2, lines 43-53 was exactly the same of the one published in the previous article in this same journal. Again, I advised the authors to rephrase the mentioned part of the text to avoid plagiarism of their own article.

-Page 2, lines 65-67, try to rephrase this sentence since the meaning is not clear.

- Page 2, lines 68-70, try to rephrase this paragraph since the meaning is not clear.

-I would suggest not splitting the introduction section in subsections (tannin based adhesives, animal glue).

-Page 2, line 72, grammar mistake to correct “and they are therefore…”

-Page 2, lines 73-75, spelling mistakes to be corrected and rephrase these sentences as they ideas are not clearly express and linked.

-The authors should correct some spelling and grammar mistakes present in Tannin-based adhesive section

-The last paragraph (lines 95-97) should be taken out the section and put in as final paragraph summarizing the purpose and aim the current work.

Materials and Methods: The majority of the experiments are properly described. Nevertheless the authors should take into consideration the following points:

-Check the spelling of title 2.2, replace “preparation” by “preparation”

-Page 3, lines 117 and 119, replace “steered” by “stirred”

-Please, indicate the stirring speed that was used in the mixing of the components in the formulation

-In section 2.3, please indicate the standard use for elaborating the panels if any.

-Line 123, correct spelling mistake in the sentence (missing final full stop).

-Explain why two replicates of each panel were only produced and not three or four. Is this answering to any standard?

Results and Discussion: On average, the results were clearly presented. Nonetheless, the authors should check and improve the resolution of all the figures and increase the size of the legends. In addition to that, please pay attention to the following points:

-In section 3.1, a strong variation in the values of MOE and MOR can be seen between the different panel samples of each resin type. Explain the origin of such a high standard deviation. I would also advise the authors considering the employment of more replicates per analysis to decrease the mentioned variation.

In this same section I was missing some explanation about the MOE/MOR results of the tannin-glued panels, which compared to the other panel types displayed the highest values.

-Page 6, lines 208-210, please explain the reported tendency. Why were those results obtained?

-In section 3.2, please explain what does the acronym TA3 mean.

In this same section, I am missing a more in detail discussion. The results were described and the tendencies were presented but it was not explained why they were that way.

-In section 3.4, please explain more clearly, what is the specific and beneficial effect of using peat moss in the panels, in terms of water absorption and thickness swell.

-In section 3.5, the discussion was too briefly presented. Please describe the benefits and differences from using each gluing material related to the physical chemical properties. In this same sense, it is necessary to point out the advantageous effect of peat moos fiber for the insulation properties.

-In section 3.6, I was missing a more specific discussion as demanded for section 3.2. It should be highlighted the benefit of each gluing agent regarding the vapor diffusion resistance.

Conclusions: The conclusions were in agreement with the results presented.

-The authors mentioned questions arising from the dependency between moisture and thermal conductivity of moss and the durability of the panels against fungi and fire. Are the authors considering about keeping on studying these parameters.

Should the previous points be reviewed and corrected, the manuscript would be meeting the standards for its publication in the present journal. Please pay special attention to plagiarism found of some parts of the current manuscript to a previous work published from the same authors.

Author Response

Reviewer 1:

Open Review

English language and style

( ) Extensive editing of English language and style required
(x) Moderate English changes required
( ) English language and style are fine/minor spell check required
( ) I don't feel qualified to judge about the English language and style

Yes

Can be improved

Must be improved

Not applicable

Does the introduction provide sufficient background and include all relevant references?

( )

(x)

( )

( )

Is the research design appropriate?

(x)

( )

( )

( )

Are the methods adequately described?

(x)

( )

( )

( )

Are the results clearly presented?

( )

(x)

( )

( )

Are the conclusions supported by the results?

(x)

( )

( )

( )

Comments and Suggestions for Authors

Dear authors,

The current work deals with a very interesting and novel topic such as the use of moss peat fibers and the possibily of incorporating tannin based glue to produce insulating panels. On average the quality of the work is good and I saw that it is the continuation of a previous work already published in this same journal during this year. However, I feel that the manuscript still present some points that should be revised and corrected before being considered for publication in this journal:

Title: No comments in this respect

Abstract: Well-structured, clear idea of the work done and within the word limitation demanded by the journal. However I detected that the half of the sentences in the abstract were the equal to those present in another work published in this same journal during this year (Production and Physical–Mechanical Characterization of Peat Moss (Sphagnum) Insulation Panels-doi.org/10.3390/ma14216601)

Previous article: Peat moss (sphagnum) is a commonly used sealant, fill, and insulation material in the past. During the efforts to rewet drained moors due to ecological considerations, the technical use of peat moss (sphagnum farming) again became the focus of attention. […] The specimens’ thermal conductivity, water vapor diffusion resistance, modulus of rupture, modulus of elasticity, internal bond, compression resistance, water absorption, and thickness swelling were measured. Physical–mechanical properties were adequate with the resin-bound panels, but not with wet process panels.

Current work: Peat moss (Sphagnum) was a commonly used sealant, fill and insulation material in the past. During the efforts to rewet drained moors due to ecological considerations, the technical use of peat moss (Sphagnum farming) is in the focus of attention again. […] The specimens’ thermal conductivity, water vapor diffusion resistance, modulus of rupture, modulus of elasticity, internal bond, compression resistance, water absorption and thickness swelling were measured. Physical-mechanical properties are adequate with the resin bound insulation 23 panels.

I strongly suggest that the authors reformulate the abstract to avoid plagiarism of their own previous article.

Answer: Thanks for the comment. The text passages have been rewritten respectively

Keywords: I would suggest replacing “insulation material” by “insulator” and “renewable materials” by “bio-based materials”

Answer:  Thanks for the suggestion. The new keywords were incorporated.

Introduction: In general, the ideas and the background of the current work were adequately presented. However, the following points should be considered:

-Same issue detected in abstract section was found here. The text presented in pages 1-2, lines 43-53 was exactly the same of the one published in the previous article in this same journal. Again, I advised the authors to rephrase the mentioned part of the text to avoid plagiarism of their own article.

Answer: Thanks for the comment. The text passages have been rewritten respectively.

-Page 2, lines 65-67, try to rephrase this sentence since the meaning is not clear.

Answer: Thanks for the comment. The text passages have been rewritten respectively.

- Page 2, lines 68-70, try to rephrase this paragraph since the meaning is not clear.

Answer: Thanks for the comment. The text passages have been rewritten respectively

-I would suggest not splitting the introduction section in subsections (tannin based adhesives, animal glue).

Answer: The introductory subsections have been merged.

-Page 2, line 72, grammar mistake to correct “and they are therefore…”

Answer: Thank you – mistake was corrected.

-Page 2, lines 73-75, spelling mistakes to be corrected and rephrase these sentences as they ideas are not clearly express and linked.

Answer: Thank you – mistake was corrected.

-The authors should correct some spelling and grammar mistakes present in Tannin-based adhesive section

Answer: Thank you – mistakes were corrected

-The last paragraph (lines 95-97) should be taken out the section and put in as final paragraph summarizing the purpose and aim the current work.

The specified paragraph was moved to the final paragraph.

Materials and Methods: The majority of the experiments are properly described. Nevertheless the authors should take into consideration the following points:

-Check the spelling of title 2.2, replace “preparation” by “preparation”

Answer: Thank you – mistake was corrected.

Page 3, lines 117 and 119, replace “steered” by “stirred” Please, indicate the stirring speed that was used in the mixing of the components in the formulation

Answer: The stirring speed has been included.

-In section 2.3, please indicate the standard use for elaborating the panels if any.

Answer: There is no standard focusing on the production of the studied panels. Therefore, the procedure used has been described in detail.

-Line 123, correct spelling mistake in the sentence (missing final full stop).

Answer: Thank you – mistake was corrected.

-Explain why two replicates of each panel were only produced and not three or four. Is this answering to any standard?

Answer: Two replicates of each panels were produced to take care of variation caused by manufacturing inhomogeneities. More replicates would of course improve the statistical power, but weren’t possible due to resource restrictions.

Results and Discussion: On average, the results were clearly presented. Nonetheless, the authors should check and improve the resolution of all the figures and increase the size of the legends. In addition to that, please pay attention to the following points:

Answer: The resolution of the figures was improved and the text size of the legends was increased.

-In section 3.1, a strong variation in the values of MOE and MOR can be seen between the different panel samples of each resin type. Explain the origin of such a high standard deviation. I would also advise the authors considering the employment of more replicates per analysis to decrease the mentioned variation.

Answer: The variation in MOE and MOR is especially high with panels with a density of 250 kg/m³ and is much lower with the lighter specimens studied. This indicates that there is some optimization potential regarding resin distribution whose effect is stronger with higher densification. Therefore, we have analyzed the data from MOR and MOE again and found an error in the data transfer from Excel and SPSS. At this point we would like to apologies for the error. However, we are very glad that you have pointed out this data shift to us.

In this same section I was missing some explanation about the MOE/MOR results of the tannin-glued panels, which compared to the other panel types displayed the highest values.

Answer: Some statistical information regarding the performance of the tannin-glued panels was added.

-Page 6, lines 208-210, please explain the reported tendency. Why were those results obtained?

Answer: Anavo was used because it allows a multivariate analysis to partial eta square values were applied in order to estimate the explanatory power of the individual variable, indecency for their significant

In section 3.2, please explain what does the acronym TA3 mean.

Answer: All acronyms are explained in Table 2. “TA3” corresponds to the tannin-bound panels, replicate number 3.

In this same section, I am missing a more in detail discussion. The results were described and the tendencies were presented but it was not explained why they were that way.

Answer: More discussion focusing on the basic trends has been added.

-In section 3.4, please explain more clearly, what is the specific and beneficial effect of using peat moss in the panels, in terms of water absorption and thickness swell.

Answer: The specific potential for using hygroscopic moss panels for capillary active interior insulation was mentioned in the text.

-In section 3.5, the discussion was too briefly presented. Please describe the benefits and differences from using each gluing material related to the physical chemical properties. In this same sense, it is necessary to point out the advantageous effect of peat moos fiber for the insulation properties.

Answer: The discussion of TC was enriched by a comparison with other insulation materials an an evaluation of the optimal panel density in this respect.

-In section 3.6, I was missing a more specific discussion as demanded for section 3.2. It should be highlighted the benefit of each gluing agent regarding the vapor diffusion resistance.

Answer: The effect of each gluing reagent was quantified and discussed in this section.

Conclusions: The conclusions were in agreement with the results presented.

-The authors mentioned questions arising from the dependency between moisture and thermal conductivity of moss and the durability of the panels against fungi and fire. Are the authors considering about keeping on studying these parameters?

Answer: Thanks for the comment. Therefore, further research has to be done.

Should the previous points be reviewed and corrected, the manuscript would be meeting the standards for its publication in the present journal. Please pay special attention to plagiarism found of some parts of the current manuscript to a previous work published from the same authors.

Reviewer 2 Report

Not sure what you mean on pg 4 related to how you broke down the peat moss..in a drum? How?  More description is needed.  Did you analyze the fiber size or particle size distribution at all?

Was the resin applied in the mixer as a drop or stream or did you air atomize the resin?  What was the resin viscosity?  Need to add these parameters as they dictate resin distribution potential.

Why did you store the resinated fibers at 10C and for how long?  Ie; why did you not just press them immediately after resin application?

Looks like you have some pretty big resin drops on your panel, that correct? (image in Figure 2).  If so, it looks like you had issues with resin distribution.  Any comments on this?

Why is the UF high density panel so variable in MOR and MOE?

How big was your density variation within your target density groups?  This should be presented to provide an assessment of panel variability of the process.  It is really hard to assess your results until you present the actual density.  I would suggest running your density as a covariate and then analyze the data.  You should change your reference of density affects to target density affects.

Figure 5, can’t see y-axis.  Figure 10 has no caption.

Author Response

Reviewer 2:

Open Review

English language and style

( ) Extensive editing of English language and style required
(x) Moderate English changes required
( ) English language and style are fine/minor spell check required
( ) I don't feel qualified to judge about the English language and style

Yes

Can be improved

Must be improved

Not applicable

Does the introduction provide sufficient background and include all relevant references?

(x)

( )

( )

( )

Is the research design appropriate?

(x)

( )

( )

( )

Are the methods adequately described?

( )

( )

(x)

( )

Are the results clearly presented?

( )

( )

(x)

( )

Are the conclusions supported by the results?

( )

(x)

( )

( )

Comments and Suggestions for Authors

Not sure what you mean on pg 4 related to how you broke down the peat moss in a drum? How?  More description is needed.  Did you analyze the fiber size or particle size distribution at all?

Answer:  The text passage was improved

Was the resin applied in the mixer as a drop or stream or did you air atomize the resin?  What was the resin viscosity?  Need to add these parameters as they dictate resin distribution potential.

Answer:  The text passage was improved

Why did you store the resinated fibers at 10C and for how long?  Ie; why did you not just press them immediately after resin application?

Answer:. To get a homogeneous gluing result the amount of three panels were mixed and glued in the drum chipper. after that the particles were stored at 10 grad C and a maximum time of 24 minutes.

Looks like you have some pretty big resin drops on your panel, that correct? (image in Figure 2).  If so, it looks like you had issues with resin distribution.  Any comments on this?

Answer: The tannin spots are due to the gluing process. The distribution of the tannin cannot be better managed with quantities in laboratory size bar. Therefore, further research has to be done.

Why is the UF high density panel so variable in MOR and MOE?

How big was your density variation within your target density groups?  This should be presented to provide an assessment of panel variability of the process.  It is really hard to assess your results until you present the actual density.  I would suggest running your density as a covariate and then analyze the data.  You should change your reference of density affects to target density affects.

Answer: The density was set as a covariate in ANOVA and the results are presented at the beginning of the paragraph. Measurement of density variation was integrated in text.

Figure 5, can’t see y-axis.  Figure 10 has no caption.

Answer: Thank you – mistake was corrected.

Submission Date

30 November 2021

Date of this review

15 Dec 2021 23:15:56

Reviewer 3 Report

The value of the manuscript is mainly cognitive. Practical or scientific values are average. However, the experimental part of the manuscritp is well designed, well performed and clearly presented. Analyses and discussion are performed properly and conclusions are adequate and come from the findings. It is almost ready for publications but some lacks and errors occur. These are:

Line 51: "sphagnum farming areas are suitable to store carbon.” I suggest add the amounts of carbon possible to store.

Lines 71 & 87: subsections 1.1. and 1.2 in Introduction are not necessary. I suggest delete them. Introduction should be undivided.

Line 102: "mass constancy” please change to „constant mass”;

Lines 117 & 119: "steered” & "steering” are incorrect. Should be „stirred” & „stirring”..

Line 123: add viscosities of the binders.

Table 1.: It would be more convenient to give amounts of components in grams. (e.g. per 100 g binder).

Line 132: Providing fraction composition of the crushed material is strongly advised.

Line 137”: "applied evenly” – How applied? Application method should be indicated.

Lines 179-180: Values should be given as "10.4±0.5%” instead of "10.4 (SD 0.5)” which is unusual.

Fig. 5 : invisible values in Y-axis.

Fig. 10: lacking figure caption.

When the above minor edits are made I suggest accept.

Author Response

Reviewer 3:

 Open Review

English language and style

( ) Extensive editing of English language and style required
( ) Moderate English changes required
( ) English language and style are fine/minor spell check required
(x) I don't feel qualified to judge about the English language and style

Yes

Can be improved

Must be improved

Not applicable

Does the introduction provide sufficient background and include all relevant references?

( )

(x)

( )

( )

Is the research design appropriate?

(x)

( )

( )

( )

Are the methods adequately described?

(x)

( )

( )

( )

Are the results clearly presented?

(x)

( )

( )

( )

Are the conclusions supported by the results?

(x)

( )

( )

( )

Comments and Suggestions for Authors

The value of the manuscript is mainly cognitive. Practical or scientific values are average. However, the experimental part of the manuscript is well designed, well performed and clearly presented. Analyses and discussion are performed properly and conclusions are adequate and come from the findings. It is almost ready for publications but some lacks and errors occur. These are:

Line 51: "sphagnum farming areas are suitable to store carbon.” I suggest add the amounts of carbon possible to store.

Answer:  The text passage was improved

Lines 71 & 87: subsections 1.1. and 1.2 in Introduction are not necessary. I suggest delete them. Introduction should be undivided.

Answer:  The text passage was improved

Line 102: "mass constancy” please change to „constant mass”;

Answer:  The sentence was improved

Lines 117 & 119: "steered” & "steering” are incorrect. Should be „stirred” & „stirring”..

Answer: Thank you – mistake was corrected.

Line 123: add viscosities of the binders.

Answer:  The text passage was improved.

Table 1.: It would be more convenient to give amounts of components in grams. (e.g. per 100 g binder).

Answer: The unit was changed to grams.

Line 132: Providing fraction composition of the crushed material is strongly advised.

Answer: The size of the crushed material (1-3 mm) was stated in the text. Unfortunately, the determination of the fraction composition was not possible due to the lack of adequate measuring equipment.

Line 137”: "applied evenly” – How applied? Application method should be indicated.

Answer:  The text passage was improved

Lines 179-180: Values should be given as "10.4±0.5%” instead of "10.4 (SD 0.5)” which is unusual.

Answer:  The values were changed

Fig. 5 : invisible values in Y-axis.

Answer: Thank you – mistake was corrected.

Fig. 10: lacking figure caption.

Answer: Thank you – mistake was corrected.

When the above minor edits are made I suggest accept.

Round 2

Reviewer 1 Report

Dear authors, I acknowledge your efforts in correcting and improving the previous version of the manuscript. Nevertheless, I still noticed that major improvements are needed regarding the discussion of the results. Should these improvements be done, the article would be able to be accepted for publication.

Best regards,

Author Response

The improvements regarding the discussion of the results are done in the new version .

Best regards Marco Morandini

Reviewer 2 Report

Responses to my comments were not complete. 

Author Response

(The authors gave the same response as above.)

Round 3

Reviewer 1 Report

Dear authors,

Considering the changes done on the manuscript, I would agree that it has a minumum level and scientific significance for its publication in this journal. Besides, you have adress most of my previous concerns.

Regards,

Author Response

Dear authors,

The current work deals with a very interesting and novel topic such as the use of moss peat fibers and the possibily of incorporating tannin based glue to produce insulating panels. On average the quality of the work is good and I saw that it is the continuation of a previous work already published in this same journal during this year. However, I feel that the manuscript still present some points that should be revised and corrected before being considered for publication in this journal:

Title: No comments in this respect

Abstract: Well-structured, clear idea of the work done and within the word limitation demanded by the journal. However I detected that the half of the sentences in the abstract were the equal to those present in another work published in this same journal during this year (Production and Physical–Mechanical Characterization of Peat Moss (Sphagnum) Insulation Panels-doi.org/10.3390/ma14216601)

.

Dear Reviewer 1:

Answer: Thanks for the comment. The text passages have been rewritten respectively

Keywords: I would suggest replacing “insulation material” by “insulator” and “renewable materials” by “bio-based materials”

Answer:  Thanks for the suggestion. The new keywords were incorporated.

Introduction: In general, the ideas and the background of the current work were adequately presented. However, the following points should be considered:

-Same issue detected in abstract section was found here. The text presented in pages 1-2, lines 43-53 was exactly the same of the one published in the previous article in this same journal. Again, I advised the authors to rephrase the mentioned part of the text to avoid plagiarism of their own article.

Answer: Thanks for the comment. The text passages have been rewritten respectively.

-Page 2, lines 65-67, try to rephrase this sentence since the meaning is not clear.

Answer: Thanks for the comment. The text passages have been rewritten respectively.

- Page 2, lines 68-70, try to rephrase this paragraph since the meaning is not clear.

Answer: Thanks for the comment. The text passages have been rewritten respectively

-I would suggest not splitting the introduction section in subsections (tannin based adhesives, animal glue).

Answer: The introductory subsections have been merged.

-Page 2, line 72, grammar mistake to correct “and they are therefore…”

Answer: Thank you – mistake was corrected.

-Page 2, lines 73-75, spelling mistakes to be corrected and rephrase these sentences as they ideas are not clearly express and linked.

Answer: Thank you – mistake was corrected.

-The authors should correct some spelling and grammar mistakes present in Tannin-based adhesive section

Answer: Thank you – mistakes were corrected

-The last paragraph (lines 95-97) should be taken out the section and put in as final paragraph summarizing the purpose and aim the current work.

The specified paragraph was moved to the final paragraph.

Materials and Methods: The majority of the experiments are properly described. Nevertheless the authors should take into consideration the following points:

-Check the spelling of title 2.2, replace “preparation” by “preparation”

Answer: Thank you – mistake was corrected.

Page 3, lines 117 and 119, replace “steered” by “stirred” Please, indicate the stirring speed that was used in the mixing of the components in the formulation

Answer: The stirring speed has been included.

-In section 2.3, please indicate the standard use for elaborating the panels if any.

Answer: There is no standard focusing on the production of the studied panels. Therefore, the procedure used has been described in detail.

-Line 123, correct spelling mistake in the sentence (missing final full stop).

Answer: Thank you – mistake was corrected.

-Explain why two replicates of each panel were only produced and not three or four. Is this answering to any standard?

Answer: Two replicates of each panels were produced to take care of variation caused by manufacturing inhomogeneities. More replicates would of course improve the statistical power, but weren’t possible due to resource restrictions.

Results and Discussion: On average, the results were clearly presented. Nonetheless, the authors should check and improve the resolution of all the figures and increase the size of the legends. In addition to that, please pay attention to the following points:

Answer: The resolution of the figures was improved and the text size of the legends was increased.

-In section 3.1, a strong variation in the values of MOE and MOR can be seen between the different panel samples of each resin type. Explain the origin of such a high standard deviation. I would also advise the authors considering the employment of more replicates per analysis to decrease the mentioned variation.

Answer: The variation in MOE and MOR is especially high with panels with a density of 250 kg/m³ and is much lower with the lighter specimens studied. This indicates that there is some optimization potential regarding resin distribution whose effect is stronger with higher densification. Therefore, we have analyzed the data from MOR and MOE again and found an error in the data transfer from Excel and SPSS. At this point we would like to apologies for the error. However, we are very glad that you have pointed out this data shift to us.

In this same section I was missing some explanation about the MOE/MOR results of the tannin-glued panels, which compared to the other panel types displayed the highest values.

Answer: Some statistical information regarding the performance of the tannin-glued panels was added.

-Page 6, lines 208-210, please explain the reported tendency. Why were those results obtained?

Answer: Anavo was used because it allows a multivariate analysis to partial eta square values were applied in order to estimate the explanatory power of the individual variable, indecency for their significant

In section 3.2, please explain what does the acronym TA3 mean.

Answer: All acronyms are explained in Table 2. “TA3” corresponds to the tannin-bound panels, replicate number 3.

In this same section, I am missing a more in detail discussion. The results were described and the tendencies were presented but it was not explained why they were that way.

Answer: More discussion focusing on the basic trends has been added.

-In section 3.4, please explain more clearly, what is the specific and beneficial effect of using peat moss in the panels, in terms of water absorption and thickness swell.

Answer: The specific potential for using hygroscopic moss panels for capillary active interior insulation was mentioned in the text.

-In section 3.5, the discussion was too briefly presented. Please describe the benefits and differences from using each gluing material related to the physical chemical properties. In this same sense, it is necessary to point out the advantageous effect of peat moos fiber for the insulation properties.

Answer: The discussion of TC was enriched by a comparison with other insulation materials an an evaluation of the optimal panel density in this respect.

-In section 3.6, I was missing a more specific discussion as demanded for section 3.2. It should be highlighted the benefit of each gluing agent regarding the vapor diffusion resistance.

Answer: The effect of each gluing reagent was quantified and discussed in this section.

Conclusions: The conclusions were in agreement with the results presented.

-The authors mentioned questions arising from the dependency between moisture and thermal conductivity of moss and the durability of the panels against fungi and fire. Are the authors considering about keeping on studying these parameters?

Answer: Thanks for the comment. Therefore, further research has to be done.

Should the previous points be reviewed and corrected, the manuscript would be meeting the standards for its publication in the present journal. Please pay special attention to plagiarism found of some parts of the current manuscript to a previous work published from